# Amaryllidaceae-Type Alkaloids from *Pancratium maritimum*: Apoptosis-Inducing Effect and Cell Cycle Arrest on Triple-Negative Breast Cancer Cells

**DOI:** 10.3390/molecules27185759

**Published:** 2022-09-06

**Authors:** Shirley A. R. Sancha, Adriana V. Gomes, Joana B. Loureiro, Lucília Saraiva, Maria José U. Ferreira

**Affiliations:** 1Faculty of Pharmacy, Research Institute for Medicines (iMed.ULisboa), Universidade de Lisboa, Av. Prof. Gama Pinto, 1649-003 Lisbon, Portugal; 2LAQV/REQUIMTE, Laboratόrio de Microbiologia, Departamento de Ciências Biolόgicas, Faculdade de Farmácia, Universidade do Porto, 4050-313 Porto, Portugal

**Keywords:** amaryllidaceae alkaloids, *Pancratium maritimum*, triple-negative breast cancer, antiproliferative effect, apoptosis, cell cycle

## Abstract

Aiming to find Amaryllidaceae alkaloids against breast cancer, including the highly aggressive triple-negative breast cancer, the phytochemical study of *Pancratium maritimum* was carried out. Several Amaryllidaceae-type alkaloids, bearing scaffolds of the haemanthamine-, homolycorine-, lycorine-, galanthamine-, and tazettine-type were isolated (**3**–**11**), along with one alkamide (**2**) and a phenolic compound (**1**). The antiproliferative effect of compounds (**1**–**11**) was evaluated by the sulforhodamine B assay against triple-negative breast cancer cell lines MDA-MB-231 and MDA-MB-468, breast cancer cells MCF-7, and the non-malignant fibroblast (HFF-1) and breast (MCF12A) cell lines. The alkaloids **3**, **5**, **7**, and **11** showed significant growth inhibitory effects against all breast cancer cell lines, with IC_50_ (half-maximal inhibitory concentration) values ranging from 0.73 to 16.3 µM. The homolycorine-type alkaloid **7** was selected for further investigation in MDA-MB-231 cells. In the annexin-V assay, compound **7** increased cell death by apoptosis, which was substantiated, in western blot analyses, by the increased expression of the pro-apoptotic protein Bax, and the decreased expression of the anti-apoptotic protein Bcl-xL. Consistently, it further stimulated mitochondrial reactive oxygen species (ROS) generation. The antiproliferative effect of compound **7** was also associated with G2/M cell cycle arrest, which was supported by an increase in the p21 protein expression levels. In MDA-MB-231 cells, compound **7** also exhibited synergistic effects with conventional chemotherapeutic drugs such as etoposide.

## 1. Introduction

Cancer is a major health concern worldwide, mainly attributed to drug resistance. Breast cancer is the most frequent cancer diagnosed among women [1]. Triple-negative breast cancer, which is well known for its therapeutic resistance, is an aggressive subtype of breast cancer with defective expression of estrogen and progesterone receptors and a lack of human epidermal growth factor receptor 2. Triple-negative breast cancer is generally associated with hereditary conditions, representing 10–15% of the newly diagnosed breast cancer [2,3]. The treatment of this type of breast cancer is mainly based on standard chemotherapy, including platinum drugs, and more recently on poly (ADP-ribose) polymerase (PARP) inhibitors. However, the use of these anticancer drugs has been limited by the occurrence of drug resistance, related to the complexity of the tumor microenvironment and heterogeneity, which is characterized by the interaction of multiple factors and signaling pathways [4,5]. Despite all efforts that have been developed in the discovery of new chemotherapeutic agents, targeting triple-negative breast cancer is still a great challenge, and there is a need for new anticancer compounds to overcome its high drug resistance [6].

To overcome multidrug resistance (MDR) in cancer, several approaches have been proposed, mainly focused on the development of molecules that can act on MDR mechanisms. The deregulation of apoptosis and cell cycle pathways are among the most frequent alterations in cancer cells responsible for drug resistance and cancer development [7]. Apoptosis is characterized by a series of biochemical events that play a key role in the regulation of many physiological and pathophysiological processes. In drug-resistant cancer cells, the mechanisms of apoptosis are suppressed, causing an imbalance between cell death and proliferation [8,9]. Moreover, changes in the cell cycle regulatory signaling pathways are commonly altered in cancer cells, which often lose their checkpoint mechanisms, contributing to cell proliferation. Therefore, targeting deficient cell checkpoints and apoptosis are considered promising strategies for improving anticancer drug development and overcoming MDR [7,10].

Natural products have been an important source for the discovery of new bioactive compounds with anticancer properties. The most known clinically used chemotherapeutic agents for cancer treatment are from natural sources [11,12]. 

Plants from the Amaryllidaceae family are characterized by containing a wide range of unique and structurally diverse bioactive Amaryllidaceae-type alkaloids. Some of them have shown great anticancer activity, mainly lycorine [13,14].

In our search for new anticancer compounds targeting multidrug resistance in cancer, e.g., [15,16,17,18,19,20,21,22,23], recently, our group has reported a derivative of a monoterpene indole alkaloid isolated from the African medicinal plant *Tabernaemontana elegans* [24] that is shown to be highly effective against breast and ovarian cancer cells [25]. Aiming at finding new anticancer compounds against breast cancer cells, this study was conducted to obtain bioactive alkaloids from *Pancratium maritimum* L. (Amaryllidaceae family), which led to the isolation of nine Amaryllidaceae-type alkaloids, along with two other compounds. 

The antiproliferative effect of compounds (**1**–**11**) was evaluated against the triple-negative breast cancer cell lines MDA-MB-231 and MDA-MB-466, the breast cancer MCF-7, and the non-malignant cell lines MCF-12A and HFF-1. Compound **7** was selected for further studies in MDA-MB-231 breast cancer cells to evaluate the mechanisms underlying its antiproliferative activity, namely the effect of the compound on cell cycle and apoptosis.

## 2. Results and Discussion

### 2.1. Isolation of Compounds

The bulbs of *P. maritimum* were exhaustively extracted with methanol. Subsequent acid-base portioning of the methanol extract, at different pH values (5 and 9) and organic solvents, yielded the alkaloid fractions, which were further fractionated by chromatographic methods, yielding several Amaryllidaceae-type alkaloids, bearing scaffolds of the haemanthamine-, homolycorine-, lycorine-, galanthamine-, and tazettine-type (**3**–**11**), namely the alkaloids haemanthidine (**3**) [13], hippeastrine (**4**) [13], lycorine (**5**) [13], 11α-hydroxygalanthamine (**6**) [26], 2α-10bα-dihydroxy-9-*O*-demethylhomolycorine (**7**) [27], epi-galanthamine (**8**) [13], 8-*O*-demethylhomolycorine (**9**) [13], tazettine (**10**) [13], and haemanthamine (**11**) [13]. The alkamide *N*-*trans*-feruloyl-tyramine (**2**) [28] and the phenolic compound 4,6-dimethoxy-2-hydroxy acetophenone (**1**) [29] were also isolated. The structures of the compounds (Figure 1) were established based on their spectroscopic data, mainly by 1D and 2D NMR and mass spectrometry (Supporting Information), which were in agreement with those reported in the literature for these compounds.

### 2.2. Biological Activity

#### 2.2.1. Antiproliferative Effect Evaluation

The antiproliferative effect of the isolated compounds (**1**–**11**) was investigated by the sulforhodamine B assay against human triple-negative breast cancer cell lines (MDA-MB-468 and MDA-MB-231) and breast ductal carcinoma cells (MCF-7) and the non-tumorigenic fibroblast (HFF-1) and breast cell lines (MCF12A). The IC_50_ values were determined after 48 h of treatment (Table 1). 

Lycorine (**5**) was found to be the most effective compound against the three breast cancer cell lines, exhibiting the lowest IC_50_ values (0.92 and 1.42 µM in the triple-negative cancer cells, MDA-MB-231 and MDA-MB-468, respectively; IC_50_ of 0.73 µM in the MCF-7 breast cancer cell line). Similarly, compounds **3** and **11** also showed strong growth inhibitory effects against MDA-MB-231 cells (4.88 and 3.95 µM, respectively), MDA-MB-468 (IC_50_ of 3.5 and 3.8 µM, respectively), and MCF-7 cell lines (IC_50_ of 2.7 and 1.6 µM, respectively). Similarly, a pronounced antiproliferative activity was also found for compound **7** against MDA-MB-231 cells (IC_50_ of 8.02 µM) and MCF-7 (IC_50_ of 6.8 µM), whereas a higher IC_50_ value of 16. A total of 3 µM was obtained in MDA-MB-468 cells. Interestingly, compound **7** displayed much higher activity than its structurally related compounds, **4** (IC_50_ of 24.70 µM) and **9** (IC_50_ > 40), suggesting that the higher number of free hydroxyl groups found in compound **7** may explain this different activity between the three compounds.

In addition, the antiproliferative effect of the compounds was evaluated on the non-malignant fibroblast (HFF-1) and breast (MCF12A) cell lines. When comparing the IC_50_ values obtained in HFF-1 cells with those obtained in cancer cell lines, a slight selectivity (SI) was found for compound **7** in MDA-MB-231 (SI of 1.65) and MCF-7 (SI of 2.37) cancer cells. Conversely, compounds **5** and **11** exhibited slight selectivity in all breast cancer cell lines in relation to MCF12A cells, particularly compound **5** against MDA-MB-231 cells (SI of 3.26) and both **5** and **11** in MCF-7 cells (SI of over 4).

In previous studies, compounds **3**, **5**, and **11** have shown antiproliferative activity against MDA-MB-231 breast cancer cells, which is in agreement with our results [30].

#### 2.2.2. Analysis of Cell Cycle, Apoptosis, and Mitochondrial ROS Generation

Compound **7**, the antiproliferative effect of which has not been investigated yet, was selected for further studies in the triple-negative breast cancer cell line MDA-MB-231.

To explore the ability of compound **7** to regulate cell cycle progression, the percentage of cells in different phases was analyzed after 24 h of exposure. As shown in Figure 2B, a two-fold IC_50_ concentration of compound **7** (16 µM) induced a significant accumulation of cells in the G2/M phase compared to the vehicle control cells. Importantly, at 16 µM, compound **7** did not significantly interfere with cell cycle, cell death, and mitochondrial ROS generation in HFF-1 cells.

#### 2.2.3. Western Blot Analyses

Once compound **7** induced cancer cell death by apoptosis, we next explored the molecular pathways involved in its apoptotic effect. For that, some apoptotic-related proteins were checked in MDA-MB-231 cells treated with 16 µM of compound **7** for 48 h (Figure 2E,F). The results showed increased levels of the pro-apoptotic protein Bax and reduced levels of the anti-apoptotic Bcl-xL, thus substantiating the data obtained in the Annexin V-FITC assay. Nonetheless, the compound did not interfere with the p53 protein expression levels (data not shown).

Moreover, in accordance with induction of a G2/M cycle arrest, compound **7** also increased the p21 protein levels. 

#### 2.2.4. Combination Therapy

To evaluate the ability of compound **7** of promoting the anticancer effect of conventional chemotherapeutic drugs, MDA-MB-231 cells were treated with a single concentration of compound **7** (at 1.5 µM; concentration with no significant effect on cancer cell growth) in combination with a DNA-damaging agent, such as etoposide. The results showed that 1.5 µM of compound **7** significantly increased the growth inhibitory effect of etoposide at the concentrations of 0.8, 1.5, and 3 µM (Figure 3). A multiple drug-effect analysis was carried out for each combination, with the calculation of the combination index (CI), which revealed synergistic effects between compound **7** and 0.8, 1.5, and 3 µM of etoposide (Table 2). Of note, for the same experimental conditions used in MDA-MB-231 cells, no synergistic effects were observed in HFF-1 cells (Table 2).

The effect of 1.5 μM of compound **7** in combination with a range of concentrations of etoposide was evaluated after 48 h of treatment using CompuSyn software to calculate the combination index (CI) values for each combined treatment: CI < 1, synergy; 1 < CI < 1.1, additive effect; CI > 1.1, antagonism.

## 3. Materials and Methods

### 3.1. General Experimental Procedure

The NMR spectra were recorded on a Bruker 300 Ultra-Shield instrument (^1^H 300 MHz, ^13^C 75 MHz). ^1^H and ^13^C chemical shifts are expressed in *δ* (ppm) referenced to the solvent used and the proton coupling constants *J* are in hertz (Hz). Optical rotations were performed using a PerkinElmer 241 polarimeter, with quartz cells of 1 dm path length. The infrared spectra were collected on an Affinity-1 (Shimadzu, Kyoto, Japan) FTIR spectrophotometer. Low-resolution mass spectrometry was conducted with a Triple Quadrupole mass spectrometer Waters AcquityTM (Waters^®^, Ireland). Column chromatography (CC) was performed on silica gel (Merck 9385, Darmstadt, Germany) or Combiflash system (teledyne-Isco; Lincolm, NE, USA), using SiO_2_ or C_18_ prepacked columns. Analytical thin-layer chromatography (TLC) was performed on pre-coated silica gel 60 F_254_ and RP-18 F_254_ plates (Merck 105,554 and 105,560, respectively), with visualization under UV light (λ 254 and 366 nm), and by spraying either with Dragendorff′s reagent or a solution of H_2_SO_4_–MeOH (1:1), followed by heating.

### 3.2. Plant Material

The whole plant of *Pancratium maritimum* was collected in August 2018, during the flowering period, at Cabo Raso, Cascais, Portugal. The plant was identified by Dr. Teresa Vasconcelos (plant taxonomist) of Instituto Superior de Agronomia, Technical University of Lisbon, Portugal. A voucher specimen (no. LISI044123) has been deposited at the herbarium João de Carvalho Vasconcellos of Instituto Superior de Agronomia.

### 3.3. Tested Compounds 

The chemical structures of compounds are presented in Figure 1. These included 4,6-dimethoxy-2-hydroxyacetophenone (**1**), *N*-trans-feruloyl-tyramine (**2**), haemanthidine (**3**), hippeastrine (**4**), lycorine (**5**), 11α-hydroxygalanthamine (**6**), 2α-10α-dihydroxy-9-*O*-demethylhomolycorine (**7**), epi-galanthamine (**8**), 9-*O*-demethylhomolycorine (**9**), tazettine (**10**), and haemanthamine (**11**). The isolation of compounds is described below. Etoposide was obtained from Sigma-Aldrich (Sintra, Portugal). All tested compounds were dissolved in dimethyl sulfoxide (DMSO) from Sigma-Aldrich (Sintra, Portugal). In all experiments, the solvent (0.1–0.25% DMSO) was included as a control.

### 3.4. Extraction and Isolation

The fresh bulbs (14.8 kg) of *P. maritimum* were sliced, dried at 40 °C for 48 h, and then minced. The resulting powder (8.19 kg) was extracted with MeOH (11 × 30 L). Afterward, the crude methanol extract (1.034 kg) was suspended in a solution of HCl 2% (*v/v*; 2.0 L), and the neutral material was removed with Et_2_O (3 × 1 L). The pH of the aqueous phase was adjusted to 5 by the addition of dilute NH_4_OH and successively extracted with CH_2_Cl_2_ (3 × 500 mL), EtOAc (4 × 1 L), and *n*-butanol (4 × 1 L), yielding, after drying over anhydrous Na_2_SO_4_ and evaporating the solvents under reduced pressure, the CH_2_Cl_2__pH5_ (13.9 g) and EtOAc_pH5_ (20 g) and the *n*-butanol_pH5_ (109 g)-soluble fractions. Then, the pH of the aqueous phase was adjusted to 9, by adding NH_4_OH, and extracted again with EtOAc (3 × 1 L) and *n*-butanol (4 × 1 L), originating two new alkaloid fractions after the above procedure (EtOAc_pH9_ 10.3 g; and *n*-butanol _pH9_ 32.2 g) (see Appendix A).

The residue of the powdered bulbs was further extracted with MeOH using a Soxhlet apparatus (500 g each dose for 10 h). The resulting MeOH extract (168 g) was submitted to acid-base extraction, as described above, yielding the EtOAc_pH9_ (3.5 g) and *n*-butanol_pH9_ (4.5 g)-soluble fractions (see Appendix A).

The alkaloid CH_2_Cl_2pH5_ (13.9g) and EtOAc_pH5_ (20g)-soluble fractions have shown the same chromatographic profile (TLC) and were combined, and then fractionated over silica gel column chromatography (*n*-hexane–EtOAc, 9:1 to 0:1; EtOAc–MeOH, 1:0 to 3:2). Column chromatography of a fraction eluted with *n*-hexane–EtOAc (4:1; 489 mg) (*n*-hexane–CH_2_Cl_2_, 1:0 to 7:3), on silica gel, yielded compound **1** (122 mg), whereas compound **2** (17 mg) was obtained by repeated column chromatography, on silica gel, of a fraction eluted with *n*-hexane–EtOAc (1:4; 3.94 g), using mixtures of CH_2_Cl_2_/MeOH (1:0 to 7:3), followed by preparative TLC (CH_2_Cl_2_/MeOH, 9:1). The alkaloid soluble fraction BuOH_pH5_ (109 g) was subjected to silica gel column chromatography, using solvent mixtures of increasing polarity (*n*-hexane–CH_2_Cl_2_, 0:1; CH_2_Cl_2_–MeOH, 1:0 to 7:3). Column chromatography of a fraction eluted with CH_2_Cl_2_–MeOH (9:1; 19.9 g) on silica gel (*n*-hexane–CH_2_Cl_2_, 1:0 to 0:1), followed by purification by Combiflash (80 g prepacked SiO2 column, RediSep^®^Rf, Teledyne Isco, Lincoln, NE, USA), using as the eluent *n*-hexane–EtOAc (1:0 to 0:1, 5% increments with an 8 mL·min^−1^), yielded compounds **3** (3.595 g) and **4** (15 mg). 

In turn, compound **5** (1.2 g) was obtained through crystallization with methanol of the crude alkaloid fraction EtOAc_pH9_ (10.3 g). Further study of the mother liquid of this fraction, by repeated column chromatography (*n*-hexane–EtOAc, 7:3 to 0:1 and EtOAc–MeOH, 0:1 to 1:0; CH_2_Cl_2_/MeOH 1:0 to 3:7) afforded compound **6** (155 mg), after further purification by crystallization (*n*-hexane–EtOAc). Crystallization of the fraction eluted with EtOAc–MeOH (7:3; 790 mg) from methanol led to the isolation of compound **7** (97 mg). The resulting mother liquid was fractionated on silica gel (*n*-hexane-EtOAc, 7:3 to 0:1; EtOAc–MeOH, 1:0 to 7:3) to yield a fraction (280 mg) that was chromatographed on aluminum oxide (CH_2_Cl_2_-MeOH, 1:0 to 1:9), followed by preparative TLC (CH_2_Cl_2_/MeOH, 19:1), affording compounds **8** (30 mg) and **9** (7 mg). 

The alkaloid soluble fraction BuOH_pH9_ (32.2 g) was fractionated over aluminum oxide (CH_2_Cl_2_–MeOH, 1:0 to 0:1) to yield several fractions. The fraction eluted with CH_2_Cl_2_–MeOH (19:1; 446 mg) was crystallized from MeOH to obtain 77 mg of the previously isolated compound **5**. The resulting mother liquid was submitted to silica gel chromatography (CH_2_Cl_2_–MeOH, 1:0 to 9:1) tof afford compound **10** (20 mg). 

Concerning the Soxhlet extract, the crystallization of EtOAc_pH9_ fraction (3.53 g) with methanol led to the isolation of compound **5** again (400 mg). Successive fractionation of the mother liquid on silica gel, first with *n*-hexane–EtOAc (1:0 to 0.1) and EtOAc–MeOH (1:0 to 7:3), and after with CH_2_Cl_2_/MeOH (1:0 to 9:1), afforded compound **11** (8 mg).

### 3.5. Human Cell Lines and Growth Conditions

Human breast adenocarcinoma MDA-MB-231, MDA-MB-468, and MCF-7 cell lines and non-tumorigenic foreskin fibroblast (HFF-1) and breast (MCF12A) cell lines were purchased from American Type Culture Collection (ATCC, Rockville, MD, USA). All cancer cell lines, except MCF12A, were routinely cultured in RPMI-1640 medium with glutamine from Corning (VWR, Carnaxide, Portugal) supplemented with 10% fetal bovine serum (FBS) from Biowest (Labclinics, Barcelona, Spain). MCF12A cells were cultured in a 1:1 mixture of Dulbecco’s modified Eagle’s medium and Ham’s F12 medium, 20 ng/mL human epidermal growth factor, 100 ng/mL cholera toxin, 0.01 mg/mL bovine insulin, and 500 ng/mL hydrocortisone and supplemented with 10% FBS. All cells were maintained in a humidified incubator at 37 °C with 5% CO_2_. All cells were routinely tested for mycoplasma contamination using the MycoAlertTM PLUS detection kit (Lonza, Basel, Switzerland).

### 3.6. Sulforhodamine B Assay

Human cell lines were seeded in 96-well plates at a density of 5.0 × 10^3^ (MDA-MB-231, MCF-7, HFF-1, and MCF12A) and 7.5 × 10^3^ (MDA-MB-468) cells/well and allowed to adhere for 24 h. Cells were treated with serial dilutions of compounds (ranging from 0.1 to 40 μM) for an additional 48 h. Effect on cell proliferation was measured by sulforhodamine B assay, as described in [33], and the half-maximal inhibitory concentration (IC_50_) was then determined for each cell line using the GraphPad Prism software version 7.0 (La Jolla, CA, USA).

### 3.7. Cell Cycle and Apoptosis Analysis

The analyses were performed basically as described in [33]. Briefly, MDA-MB-231 cells were seeded in 6-well plates at a density of 1.5× 10^5^ cells/well for 24 h, followed by treatment with 16 µM of compound **7** for an additional 48 h. For cell cycle analysis, cells were stained with propidium iodide (PI; Sigma-Aldrich, St. Louis, MO, USA). The cells were then analyzed by flow cytometry, and cell cycle phases were identified and quantified using the FlowJo × 10.0.7 Software (Treestar, Ashland, OR, USA). For apoptosis, cells were stained using the Annexin V-FITC Apoptosis Detection Kit I from BD Biosciences (Enzifarma, Porto, Portugal), according to the manufacturer’s instructions. The AccuriTM C6 flow cytometer and the BD Accuri C6 software (BD Biosciences) were used.

### 3.8. Mitochondrial Reactive Oxygen Species (ROS) Generation 

For mitochondrial ROS generation assessment, MDA-MB-231 cells were seeded in 6-well plates at a density of 1.5 × 10^5^ cells/well for 24 h, followed by treatment with 16 µM of compound **7** for an additional 24 h. Thereafter, cells were incubated with 3 μM MitoSOX (Thermo Fisher Scientific, Invitrogen, Portugal) for 30 min at 37 °C and analyzed by flow cytometry.

### 3.9. Western Blot Analysis

MDA-MB-231 cells were seeded in 6-well plates at a density of 1.5 × 10^5^ cells/well for 24 h, followed by treatment with 16 µM of compound **7**. Protein extracts were quantified using the Bradford reagent (Sigma-Aldrich). Proteins were run in SDS-PAGE and transferred to a Whatman nitrocellulose membrane from Protan (VWR, Carnaxide, Portugal). After blocking, proteins were identified using specific primary antibodies: rabbit polyclonal anti-p21 (Santa Cruz Biotechnology, Dallas, DX, USA), rabbit polyclonal anti-Bax, and rabbit monoclonal anti-Bcl-xL (Invitrogen, Waltham, MA, USA), followed by the respective anti-mouse/anti-rabbit HRP-conjugated secondary antibodies (Santa Cruz Biotechnology). GAPDH was used as a loading control. The signal was detected with the ECL Amersham kit from GE Healthcare (VWR, Carnaxide, Portugal). Two detection methods were used: the Kodak GBX developer and fixer (Sigma-Aldrich) or the ChemiDoc™ XRS Imaging System from Bio-Rad Laboratories (Amadora, Portugal). 

### 3.10. Combination Therapy Assays

To assess the synergistic effect of compound **7** with the conventional chemotherapeutic drug etoposide, MDA-MB-231 cells were treated with compound 7 at 1.5 µM (concentration that did not interfere with the growth of cancer cells) and/or increasing concentrations of etoposide, for 48 h. The effect of combined treatments on cell proliferation was analyzed by SRB assay. For each combination, the combination index (CI) values were calculated using the CompuSyn Software version 1.0; CI values < 1, 1 < CI < 1.1, and >1.1 indicate synergistic, additive, and antagonistic effects, respectively [34]. 

### 3.11. Statistical Analysis

The data were analyzed statistically using the GraphPad Prism (La Jolla, CA, USA; version 8.0.2) software for Microsoft Windows 10. For the comparison of the two groups, the unpaired Student’s *t*-test was used. For the comparison of multiple groups, the two-way ANOVA followed by Sidak’s test was used. Statistical significance was set as * *p* < 0.05.

## 4. Conclusions

To summarize, among the nine Amaryllidaceae-type alkaloids isolated from *P. maritimum,* bearing different scaffolds, the most significant growth inhibitory activity against different human breast cancer cell lines was found for compounds **3**, **5**, **7,** and **11**. When comparing the IC_50_ values obtained for compound **7** with those of **4** and **9**, sharing the same homolycorine-type scaffold, the higher number of hydroxyl groups in compound **7** seems to be associated with its higher activity. Further investigation of the mechanism of action of compound **7** against the triple-negative human breast cancer MDA-MB-231 cell line revealed that its antiproliferative effect was associated with the induction of apoptosis and cell cycle arrest through a p53-independent pathway. Interestingly, compound **7** seems to interfere with the expression levels of proteins of the Bcl-2 family, namely increasing Bax and reducing Bcl-xL, which may indicate a potential involvement of the mitochondrial pathway in its antitumor activity. In fact, an increase in mitochondrial ROS production was also observed upon compound **7** treatment. The results obtained in this work also revealed that triple-negative breast cancer therapy might greatly benefit from the combination of etoposide with compound **7**.

## Figures and Tables

**Figure 1 molecules-27-05759-f001:**
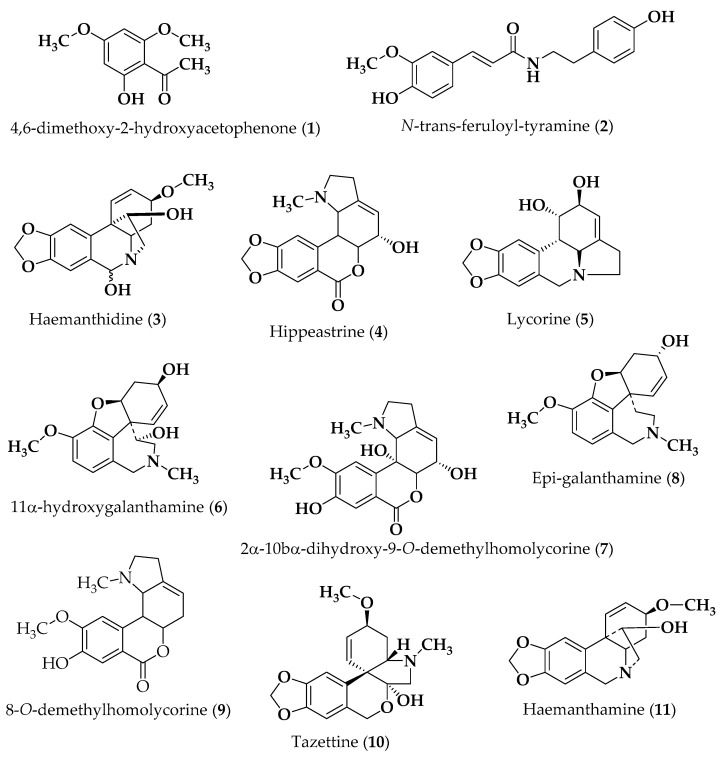
Chemical structures of compounds **1**–**11**.

**Figure 2 molecules-27-05759-f002:**
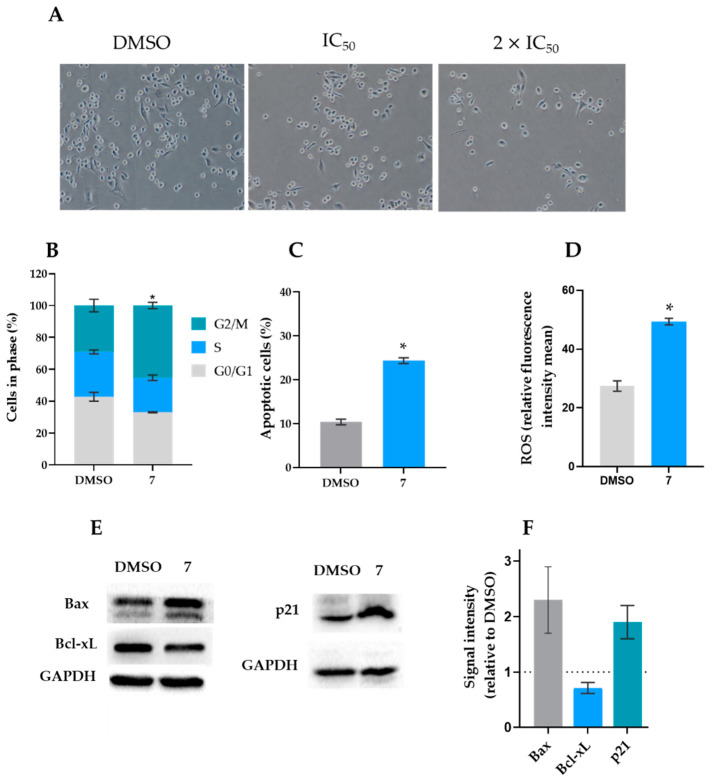
Compound **7** induced cell cycle arrest, apoptosis, and ROS production, in MDA-MB-231 cells. (**A**) Effect of 8 (IC_50_) and 16 (2 × IC_50_) µM of compound **7** on MDA-MB-231 cell cytotoxicity after 48 h of treatment. (**B**) Effect of 16 µM of compound **7** on cell cycle progression after 48 h of treatment in MDA-MB-231 cells; the percentage of cells in each phase of the cell cycle was analyzed by flow cytometry, using PI; data are mean ± SEM of three independent experiments. Values are significantly different from DMSO (* *p* < 0.05, unpaired Student’s *t*-test). (**C**) Effect of 16 µM of compound 7 on apoptosis after 48 h of treatment; the percentage of Annexin-positive cells was analyzed by flow cytometry using PI and Annexin-V staining; data are mean ± SEM of three independent experiments. Values are significantly different from DMSO (* *p* < 0.05, unpaired Student’s *t*-test). (**D**) Effect of 16 µM of compound **7** on mitochondrial ROS generation after 24 h of treatment; data are mean ± SEM of three independent experiments (* *p* < 0.05, unpaired Student’s *t*-test). (**E**) Effect of 16 µM of compound **7** after 48 h of treatment on the protein expression levels in MDA-MB-231 cells. Immunoblots are representative of three independent experiments. GAPDH was used as a loading control. (**F**) The graph represents the quantification of protein expression levels relative to DMSO; data are mean ± SD of three independent experiments. The ability of compound **7** to induce apoptosis in this cell line was also assessed by cytometry, using the Annexin V-FITC assay after 48 h of treatment. Apoptosis is characterized by distinct morphologic features. In the early stages of apoptosis, the membrane phospholipid phosphatidylserine is translocated from the inner to the outer plasma membrane leaflet. Annexin V is a phospholipid-binding protein with a high affinity to phosphatidylserine, which is able to detect apoptotic cells with the externalization of phosphatidylserine [32]. The results showed a significant increase in the percentage of apoptotic cells in MDA-MB-231 cells compared to the vehicle control after treatment with 16 µM of compound **7** (Figure 2C). In line with this, it was further observed that 16 µM of compound **7** increased ROS production in MDA-MB-231 cells (Figure 2D).

**Figure 3 molecules-27-05759-f003:**
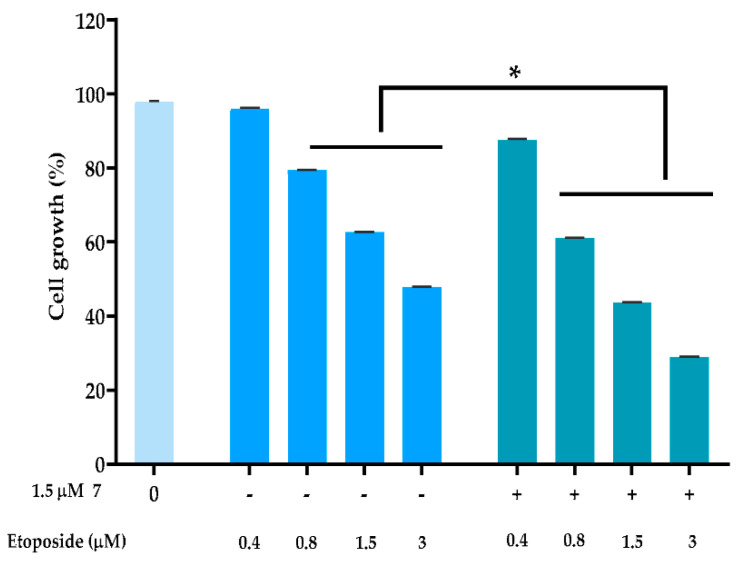
Compound **7** sensitized MDA-MB-231 cells to the effect of etoposide. Cells were treated with a range of concentrations of the conventional chemotherapeutic drug alone and in combination with 1.5 μM of compound **7**; cell proliferation was measured after 48 h of treatment; the growth obtained with control (DMSO) was set as 100%; data are mean ± SEM; values significantly different from chemotherapeutic drug alone: * *p* < 0.05 (two-way ANOVA followed by Sidak’s test).

**Table 1 molecules-27-05759-t001:** Effect of compounds **1**–**11** on the growth in a panel of human breast cancer (triple negative MDA-MB-231 and MDA-MB-466 cell lines, and MCF-7 cells) and non-tumorigenic fibroblast (HFF-1 and breast (MCF12A)) cell lines.

Compounds/ Cell Line	IC_50_ (µM) ^a^					SI ^b^					
	MDA-MDB-231 (A)	MDA-MB-468 (B)	MCF-7 (C)	MCF12A (D)	HFF-1 (E)	D/A	D/B	D/C	E/A	E/B	E/C
**1**	>40	-	-	-	-	-	-	-	-	-	-
**2**	>40	-	-	-	-	-	-	-	-	-	-
**3**	4.9 ± 1.16	3.5 ± 0.3	2.7 ± 0.1	5.0 ± 0.5	2.7 ± 0.15	** *1.02* **	** *1.42* **	** *1.85* **	0.56	0.78	1.01
**4**	24.7 ± 4.30	-	-	-	-	-	-	-	-	-	-
**5**	0.9 ± 0.20	1.4 ± 0.01	0.7 ± 0.01	3.0 ± 0.3	1.3 ± 0.30	**3.26**	** *2.11* **	**4.10**	** *1.41* **	0.91	** *1.78* **
**6**	>40	-	-	-	-	-	-	-	-	-	-
**7**	8.0 ± 1.68	16.3 ± 1.3	6.8 ± 0.2	5.6 ± 0.8	13.3 ± 3.28	0.69	0.34	0.82	** *1.65* **	0.81	** *2.37* **
**8**	>40	-	-	-	-	-	-	-	-	-	-
**9**	>40	-	-	-	-	-	-	-	-	-	-
**10**	39.0 ± 1.41	-	-	-	-	-	-	-	-	-	-
**11**	3.9 ± 0.97	3.8 ± 0.4	1.6 ± 0.1	6.5 ± 1.1	2.7 ± 0.72	** *1.64* **	** *1.71* **	**4.06**	0.68	0.71	1.7
**Etoposide**	2.9 ± 0.1	1.7 ± 0.05	3.1 ± 0.05	2.3 ± 0.13	3.8 ± 0.09	*0.79*	*1.35*	0.74	*1.31*	*2.23*	1.23

^a^ IC_50_ values were determined by sulforhodamine B assay after 48 h of treatment (growth obtained with the vehicle was set as 100%). Data are mean ± SEM of four to six independent experiments. ^b^ SI: Selectivity index; SI < 1 values denote lack of selectivity, 1 < SI < 3 mean slight selectivity and are marked with bold and italics; 3 < SI < 6 values indicate moderate selectivity and are highlighted in bold, whereas values of SI > 6 indicate that compounds are strongly selective [31]. Etoposide was used as a positive control.

**Table 2 molecules-27-05759-t002:** Effect of compound **7** in combination with etoposide, in MDA-MB-231 and HFF-1 cells.

Combination with Etoposide (μM)	Combination Index	
	MDA-MB-231	HFF-1
0.4	1.23	1.19
0.8	0.87	1.34
1.5	0.81	1.12
3	0.86	1.09

## Data Availability

The data that support the findings of this study are available from the corresponding author upon request.

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
