# Peer review of "Amaryllidaceae-Type Alkaloids from *Pancratium maritimum*: Apoptosis-Inducing Effect and Cell Cycle Arrest on Triple-Negative Breast Cancer Cells"

_molecules, 2022, doi:10.3390/molecules27185759_

Round 1
Reviewer 1 Report
The manuscript described the apoptosis-inducing effect and 2 cell cycle arrest on triple-negative breast cancer cells of amaryllidaceae-type alkaloids from Amaryllidaceae plant Pancratium maritimum.
The manuscript might not be considered to be published in Molecules in recent version. The manuscript might only be accepted after major revision. Please find below some suggestions and comments:
1. What is the rationale for using etoposide as a drug to test the synergistic effects with Amaryllidaceae-type alkaloid, compound 7? Etoposide is a chemotherapy medication used for the treatments of a number of types of cancer, including testicular cancer, lung cancer, lymphoma, leukemia, neuroblastoma, and ovarian cancer. Etoposide is not used against breast cancer. The aim for the research was to find new anti-cancer compounds against breast cancer cells.
2. The structures of compounds were established based on the 1D, 2D NMR and MS spectroscopic data. The authors provided the data in supporting information, and compared it with the reported data in the literature. However, the references were not cited in the main text nor in supporting information.
3. The specific rotation of some compounds was reported greater than ±180. For example, [?] 25 ? = + 680.7 (c, 0.092, CHCl3) of 8-O-demethylhomolycorine (9) was reported. What are the true values for these compounds?
4. The NMR spectra of some compounds were measuring by mixing D-solvents of CDCl3 and MeOH-d4. What was the mixing ratio?
5. The manuscript should be carefully edited. Something was missed after Some of … in line 64.
Author Response
Manuscript ID: molecules-1862240
Title: Amaryllidaceae-type alkaloids: apoptosis-inducing effect and cell cycle arrest on triple-negative breast cancer cells
Response to the referees' comments
We are thankful to reviewers for their constructive comments and we have revised the manuscript accordingly.
Reviewer #1
- What is the rationale for using etoposide as a drug to test the synergistic effects with Amaryllidaceae-type alkaloid, compound 7? Etoposide is a chemotherapy medication used for the treatments of a number of types of cancer, including testicular cancer, lung cancer, lymphoma, leukemia, neuroblastoma, and ovarian cancer. Etoposide is not used against breast cancer. The aim for the research was to find new anti-cancer compounds against breast cancer cells.
Author’s response: The idea was to analyze the potential of compound 7 to sensitize tumor cells to the effect of DNA-damaging agents such as etoposide. Although not so widely used as in the types of cancer mentioned by the Reviewer, etoposide is also used in breast cancer, particularly metastatic breast cancer by oral administration and in combination with other therapeutics (see for example https://www.frontiersin.org/articles/10.3389/fonc.2020.565384/full;
https://pubmed.ncbi.nlm.nih.gov/29413403/; https://doi.org/10.1016/j.annonc.2021.03.127). This explanation was included in the manuscript: “…in combination with a DNA-damaging agent such as etoposide, which combined with other therapeutics has been explored mainly in metastatic breast cancer [33–35]”.
- 2. The structures of compounds were established based on the 1D, 2D NMR, and MS spectroscopic data. The authors provided the data in supporting information and compared it with the reported data in the literature. However, the references were not cited in the main text nor in supporting information.
Author’s response: In the original manuscript we have cited a book chapter, from Bastida J. et al, 2006, entitled “Chemical, and Biological Aspects of Narcissus Alkaloids “. This article reports the NMR data for most of the compounds. Anyway, we agree that the reference was not clearly indicated and, in the revised version, we have repeated it for each compound.
- The specific rotation of some compounds was reported greater than ±180. For example, [?] 25 ?= + 680.7 (c, 0.092, CHCl3) of 8-O-demethylhomolycorine (9) was reported. What are the true values for these compounds?
Author’s response: We apologize for this mistake. The values were corrected (supporting information) and were compared to those described in the literature for each compound, adding also the corresponding references.
- The NMR spectra of some compounds were measuring by mixing D-solvents of CDCl3 and MeOH-d4. What was the mixing ratio?
Author’s response: The corresponding mixing ratio (4:1) was added
- The manuscript should be carefully edited. Something was missed after Some of … in line 64.
Author’s response: We apologize for this editing misstep. The missing part was now correctly added.

Reviewer 2 Report
The authors discuss the anti-tumor effects of different extracts obtained from Pancratium maritimum on different tumor cell lines MDA-MB-231, MDA-MB-468, MCF-7, MCF12A and a non-tumor cell line HFF-1.
If from a chemical point of view the manuscript does not present important criticalities, from a biological point of view there are some aspects that are not clear.
1) The authors report data on IC50, however, the authors should provide more details on the toxicity of all extracts. Pictures of the health of the cells would be very important. In this regard, see and cite Figure 3 in Gupta, A.K. et al. Artocarpus lakoocha Roxb. and Artocarpus heterophyllus Lam. Flowers: New Sources of Bioactive Compounds. Plants 2020, 9, 1329. https://doi.org/10.3390/plants9101329.
2) It is unclear whether the analyzed extracts are toxic to HFF-1 cells. If a compound is toxic to all cells it is difficult to find its usefulness in the clinical setting.
3) The data in figure 2 are incomplete. The authors show only the biological response on MDA-MB-231 cells, however, I believe it is essential to observe the data on control cells (HFF-1).
4) Regarding the 2D figure again, the authors must insert a graph that analyzes the variation of expression of the different markers. Showing only expression bands is not acceptable.
5) The data in Figure 3 must also show the effects on HFF-1 control cells.
6) In the introduction together with reference 8 I would also add: Abate, G. et al .. Phytochemical Analysis and Anti-Inflammatory Activity of Different Ethanolic Phyto-Extracts of Artemisia annua L. Biomolecules 2021, 11, 975. https: // doi .org / 10.3390 / biom11070975
7) The title is ambiguous, does the manuscript refer to extracts from several Amaryllidaceae or only from Pancratium maritimum?
Author Response
Reviewer #2
- The authors report data on IC50, however, the authors should provide more details on the toxicity of all extracts. Pictures of the health of the cells would be very important. In this regard, see and cite Figure 3 in Gupta, A.K. et al. Artocarpus lakoocha Roxb. and Artocarpus heterophyllus Lam. Flowers: New Sources of Bioactive Compounds. Plants 2020, 9, 1329. https://doi.org/10.3390/plants9101329.
Author’s response: As suggested by the reviewer, pictures of the MDA-MB-231 cells treated with solvent (DMSO) and 8 and 16 µM of compound 7 (IC50 and 2-fold IC50) were included in Figure 2 (panel A).
- It is unclear whether the analyzed extracts are toxic to HFF-1 cells. If a compound is toxic to all cells it is difficult to find its usefulness in the clinical setting.
Author’s response: We understand and agree with the reviewer’s comment. When comparing the IC50 values obtained in non-malignant fibroblast (HFF-1) and breast (MCF12A) cell lines with those obtained in cancer cell lines, a slight selectivity (SI) was found for some compounds, as referred to in section 2.2.1. These data may contradict the nonspecific cytotoxicity of these compounds.
- The data in figure 2 are incomplete. The authors show only the biological response on MDA-MB-231 cells, however, I believe it is essential to observe the data on control cells (HFF-1).
Author’s response: In fact, at 16 µM (IC50 of compound 7 in MDA-MB-231 cells), compound 7 did not induce significant cell cycle arrest, apoptosis, and ROS generation, in HFF-1 cells. As suggested by the Reviewer, these data were included in the revised manuscript: “Importantly, at 16 µM, compound 7 did not significantly interfere with cell cycle, cell death, and mitochondrial ROS generation, in HFF-1 cells”.
- Regarding the 2D figure again, the authors must insert a graph that analyzes the variation of expression of the different markers. Showing only expression bands is not acceptable.
Author’s response: As suggested by the Reviewer, the quantification of protein expression levels relative to DMSO (mean of three independent experiments) was included in Figure 2 (panel F in the revised manuscript).
- The data in Figure 3 must also show the effects on HFF-1 control cells.
Authors' response: Considering the Reviewer’s suggestion, the Table 2 was included in the revised manuscript to supplement Figure 3. With this Table 2, it is shown that for the same experimental conditions used in MDA-MB-231 cells, no synergistic effects were observed in HFF-1 cells. This information was also included in the text.
Table 2. Effect of compound 7 in combination with etoposide, in MDA-MB-231 and HFF-1 cells.
|
Combination with etoposide (mM) |
Combination index |
|
|
MDA-MB-231 |
HFF-1 |
|
|
0.4 |
1.23 |
1.19 |
|
0.8 |
0.87 |
1.34 |
|
1.5 |
0.81 |
1.12 |
|
3 |
0.86 |
1.09 |
Effect of 1.5 mM of compound 7 in combination with a range of concentrations of etoposide was evaluated, after 48 h of treatment, using CompuSyn software to calculate combination index (CI) values for each combined treatment: CI < 1, synergy; 1 < CI < 1.1, additive effect; CI > 1.1, antagonism.
6) In the introduction together with reference 8 I would also add: Abate, G. et al .. Phytochemical Analysis and Anti-Inflammatory Activity of Different Ethanolic Phyto-Extracts of Artemisia annua L. Biomolecules 2021, 11, 975. https: // doi .org / 10.3390 / biom11070975
Author’s response: We appreciate the suggestion but instead this reference, which reports the anti-inflammatory activity of extracts, we have included the reference below that we think it is more appropriate:
Agarwal, G.; Carcache, P.J.B.; Addo, E.M.; Kinghorn, A.D. Current Status and Contemporary Approaches to the Discovery of Antitumor Agents from Higher Plants. Biotechnol. Adv. 2020, 38, 107337.
7) The title is ambiguous, does the manuscript refer to extracts from several Amaryllidaceae or only from Pancratium maritimum?
Author´s response: We agree with the reviewer’s comment and have changed the title accordingly. The new title is: “Amaryllidaceae-type alkaloids from Pancratium maritimum: apoptosis-inducing effect and cell cycle arrest on triple-negative breast cancer cells”

Round 2
Reviewer 1 Report
This manuscript is a revised version for the manuscript of “Amaryllidaceae-type alkaloids from Pancratium maritimum: apoptosis-inducing effect and cell cycle arrest on triple-negative breast cancer cells.” The authors have responded to all comments from pre-reviewers and made revision. Only one logical problem is needed to be reconsidered.
The rationale is not convinced for using etoposide as a drug to test the synergistic effects with Amaryllidaceae-type alkaloid, compound 7. The authors listed 3 references regarding to etoposide, a DNA-damaging agent, which combined with other therapeutics has been explored mainly in metastatic breast cancer. The drugs, trastuzumab and/or taxane, are used to treat patients with HER2-positive metastatic breast cancer. The etoposide is used as a combined chemotherapeutic drug for preclinical studies of new or old anticancer drugs. The studies showed some beneficial effects of the combination with etoposide. Therefore, etoposide is still not a chemotherapeutic drug for the treatment of metastatic breast cancer. The idea, a tyrosine kinase inhibitor and a DNA-damaging agent, might have some beneficial effects for treating metastatic breast cancer. What is the rationale for this study?
The reviewer suggested the manuscript be considered to be published in Molecules after minor revision.
Author Response
Manuscript ID: molecules-1862240
Title: Amaryllidaceae-type alkaloids: apoptosis-inducing effect and cell cycle arrest on triple-negative breast cancer cells
Response to the referees' comments
Reviewer #1
This manuscript is a revised version for the manuscript of “Amaryllidaceae-type alkaloids from Pancratium maritimum: apoptosis-inducing effect and cell cycle arrest on triple-negative breast cancer cells.” The authors have responded to all comments from pre-reviewers and made revision. Only one logical problem is needed to be reconsidered.
The rationale is not convinced for using etoposide as a drug to test the synergistic effects with Amaryllidaceae-type alkaloid, compound 7. The authors listed 3 references regarding to etoposide, a DNA-damaging agent, which combined with other therapeutics has been explored mainly in metastatic breast cancer. The drugs, trastuzumab and/or taxane, are used to treat patients with HER2-positive metastatic breast cancer. The etoposide is used as a combined chemotherapeutic drug for preclinical studies of new or old anticancer drugs. The studies showed some beneficial effects of the combination with etoposide. Therefore, etoposide is still not a chemotherapeutic drug for the treatment of metastatic breast cancer. The idea, a tyrosine kinase inhibitor and a DNA-damaging agent, might have some beneficial effects for treating metastatic breast cancer. What is the rationale for this study?
Author’s response: Although we knew that etoposide would not be one of the most used chemotherapeutic drugs in the treatment of breast cancer, it is a DNA-damage agent successfully used, in combination therapy, for the treatment of several types of cancer. In fact, ongoing clinical trials and preclinical studies with this compound against metastatic breast cancer reinforce these data. Additionally, we had the information on its clinical use for metastatic breast cancer treatment (see, for example, Chemotherapy for Metastatic Breast Cancer at https://www.mskcc.org/cancer-care/types/breast/treatment/systemic-therapy/chemotherapy#anchor-2). Despite this, we understand the Reviewer's concern, so the 3 references were removed from the article, justifying the use of etoposide as a way of evaluating the potential of our compound in combination with DNA-damage agents, namely for breast cancer treatment."
Reviewer 2 Report
The authors have excellently reviewed the manuscript. In my opinion, the manuscript can be published
Author Response
Reviewer #2
The authors have excellently reviewed the manuscript. In my opinion, the manuscript can be published
Author’s response: We thank again the reviewer for the careful revision of the manuscript.